# Factors Affecting Treatment with Life-Saving Interventions, Computed Tomography Scans and Specialist Consultations

**DOI:** 10.3390/ijerph17082914

**Published:** 2020-04-23

**Authors:** Chu-Chieh Chen, Chin-Yi Chen, Ming-Chung Ko, Yi-Chun Chien, Emily Chia-Yu Su, Yi-Tui Chen

**Affiliations:** 1Department of Health Care Management, National Taipei University of Nursing and Health Sciences, Taipei City 108, Taiwan; 2Auditing and Advising Division, Trust Association of Republic of China, Taipei City 106, Taiwan; 3Department of Urology, Taipei City Hospital, Taipei City 103, Taiwan; 4School of Medicine, Fu-Jen Catholic University, New Taipei City 242, Taiwan; 5Department of Otorhinolaryngology, China Medical University Hospital, Taichung City 404, Taiwan; 6Graduate Institute of Biomedical Informatics, Taipei Medical University, Taipei City 110, Taiwan; 7Clinical Big Data Research Center, Taipei Medical University Hospital, Taipei City 110, Taiwan

**Keywords:** immediate, life-saving intervention, computed tomography scans, specialist consultation, triage and acuity scale

## Abstract

*Background:* Emergency treatments determined by emergency physicians may affect mortality and patient satisfaction. This paper attempts to examine the impact of patient characteristics, health status, the accredited level of hospitals, and triaged levels on the following emergency treatments: immediate life-saving interventions (LSIs), computed tomography (CT) scans, and specialist consultations (SCs). *Methods:* A multivariate logistic regression model was employed to analyze the impact of patient characteristics, including sex, age, income and the urbanization degree of the patient’s residence; patient health status, including records of hospitalization and the number of instances of ambulatory care in the previous year; the Charlson Comorbidity Index (CCI) score; the accredited level of hospitals; and the triaged level of emergency treatments. *Results:* All the patient characteristics were found to impact receiving LSI, CT and SC, except for income. Furthermore, a better health status was associated with a decreased probability of receiving LSI, CT and SC, but the number of instances of ambulatory care was not found to have a significant impact on receiving CT or SC. This study also found no evidence to support impact of CCI on SC. Hospitals with higher accredited levels were associated with a greater chance of patients receiving emergency treatments of LSI, CT and SC. A higher assigned severity (lower triaged level) led to an increased probability of receiving CT and SC. In terms of LSI, patients assigned to level 4 were found to have a lower chance of treatment than those assigned to level 5. *Conclusions:* This study found that several patient characteristics, patient health status, the accredited level of medical institutions and the triaged level, were associated with a higher likelihood of receiving emergency treatments. This study suggests that the inequality of medical resources among medical institutions with different accredited levels may yield a crowding-out effect.

## 1. Introduction

The emergency departments (EDs) of medical institutions represent a unique way for the public to enter the health care system. EDs provide 24-h medical services throughout the year [1,2]. As soon as patients arrive to the emergency department (ED), they are examined by a triage registered nurse who rapidly conducts a brief, focused assessment and assigns different levels of acuity according to the triage systems [3]. The major function of a triage system is to determine the degree of urgency of the disease and to prioritize the order of treatment. ED treatment may vary across different types of disease, such as potential stroke, trauma, and myocardial infarction. In general, the appropriate treatment should be determined and provided to emergency patients based on the ED physician’s judgment [4].

The determination of emergency treatments may affect outcomes such as patient mortality, the cost and quality of care for hospitals and the effective use of medical resources by medical administration authorities [4]. A delay in definitive ED treatments may increase mortality rates. For example, several studies suggest that a delay in percutaneous coronary intervention for myocardial infarction patients may yield an adverse impact on heart muscle and increase mortality rates [5,6]. Thus, information about the factors affecting emergency treatments is very important to avoid medical errors and the overuse of medical resources. This study intends to answer whether patient characteristics, including sex, age, income, and urbanization degree of residence, affect the probability of receiving ED treatments, including life-saving interventions (LSIs), computed tomography (CT) scans and specialist consultations (SCs). Patient health status, including records of hospitalization and ambulatory care in the previous year, and CCI score were also examined to determine their association with the choice of treatment. Furthermore, the accredited level of hospitals may affect the provision of ED treatments due to the availability of medical resources [7]. Thus, the association between the accredited level of medical institutions and emergency treatments was also analyzed in this study. Finally, the assigned acuity (the triaged level) also plays an important role in the determination of emergency treatments. In this study, emergency treatments were considered dependent variables, while the triaged level and other factors served as independent variables.

## 2. Research Methods

This study was based on a cross-sectional study to examine the factors affecting emergency treatments for ED patients. A multivariate logistic regression model was employed to analyze the factors affecting ED treatments. In this study, ED treatments included immediate LSIs, CT scans, and SCs.

Immediate LSI has been widely performed in the ED and includes cardiopulmonary resuscitation (CPR), endotracheal tube insertion, noninvasive positive pressure, electrical defibrillation or cardioversion, transcutaneous pacemakers, electrocardiograms monitoring, thoracocentesis, pericardial puncture, chest intubation, central venous pressure catheter intubation, peripheral arterial line insertion, and general blood transfusions. For example, rapid sequence intubation is widely used for airway control in EDs [8]. The use of CT in EDs is also popular worldwide [9,10,11]. For example, previous studies suggest that patients with clinical suspicion of acute mesenteric ischemia should be tested by CT due to the effectiveness of CT for detecting this outcome [12,13,14]. In addition, CT included both computed tomography without contrast and computed tomography with contrast in this paper. Basically, SC involves ED physicians who request assistance from other specialists. The performance of SC is essential for the improvement of medical care and the reduction in overcrowding [15,16].

The determination of emergency treatments may depend on a complex interplay of the care providers (hospital levels) and the acuity of the claimed disease as well as patient characteristics [17]. Thus, emergency treatment served as a dependent variable, while triaged levels were employed as independent variables. Some control variables, including patient characteristics, patient health status, and the accredited levels of hospitals, were also incorporated into the research structure, as shown in Figure 1. The statistical analysis was conducted using SPSS 16 software, and a P-value of < 0.05 was considered statistically significant.

In general, patient characteristics such as sex, age, income, and the urbanization degree of residence play important roles in the utilization of emergency services [4,18]. Age was divided into three groups: (1) aged 0–18, (2) aged 19–64, and (3) aged ≥ 65. Income was classified into: (1) low-income households and (2) normal households. According to Taiwan’s regulations, a household may be approved as a low-income household by local municipalities if their average monthly income in the household falls below the lowest living index that is based on the living standard and defined by the government. For example, the minimum income to support living standards in New Taipei City was NT$ 11,800 (US$ 390) per month for each individual in the family in 2012 [19]. Thus, households with an average income of less than US$ 390 per month per individual were classified as low-income households in New Taipei City in 2012. In Taiwan, the administration structure of government is divided into three levels: municipal cities, counties and second-level cities, and third-level cities and towns. Municipal cities are more developed with higher population densities, followed by second-level units and third-level units. In this paper, the urbanization degree of residence was categorized into three groups: municipal cities, satellite cities, and villages. As cities administered directly by counties generally have higher population densities than other towns, second-level cities and county-administered cities were considered satellite cities. All villages in towns under counties, excluding county-administered cities, were categorized as villages.

The variables of patient health status included hospitalization in the previous year, records of ambulatory care in the previous year, and the CCI. As the average number of instances of ambulatory care was 14 times per year in recent years [20], a patient who was hospitalized or received ambulatory care more than 15 times in the previous year was seen as having a worse health status. In this study, the CCI was employed to assess the comorbidity of patients [21] by using the measurement proposed by Deyo et al. [22]. The disease category was defined by the diagnosis or treatment code of ICD-9-CM (The International Classification of Diseases, 9th Revision, Clinical Modification). Different weighting scores were given: one point for myocardial infarct, congestive heart failure, peripheral vascular disease, cerebrovascular disease, dementia, chronic pulmonary disease, rheumatologic disease, ulcer disease, mild liver disease and diabetes; two points for hemiplegia or paraplegia, renal disease, diabetes with chronic complications, and any malignancy including leukemia and lymphoma; three points for moderate or severe liver disease; and six points for metastatic solid tumor and AIDS. By consulting with several studies (e.g., [23,24,25]), patient comorbidity was divided into three groups based on the CCI scores: (1) score 0, (2) score 1–2, and (3) score ≥ 3.

In Taiwan, medical institutions are categorized into four levels: academic medical centers, regional hospitals, district hospitals, and clinics based on accreditation standards. Clinics are only responsible for ambulatory care and referral to hospitals for in-need patients. Thus, no data in association with clinics were available in this study.

All patients presenting to the ED at first were triaged upon arrival, based on the vital signs and their chief complaint. The senior medical staff in the emergency department was responsible for the triage. incorporated with a computer-aided decision-making system according to the classification criteria of the Taiwan Triage and Acuity Scale (TTAS). The degree of urgency based on the TTAS consists of five levels: resuscitation (level 1), emergent (level 2), urgent (level 3), less urgent (level 4) and nonurgent (level 5).

In this paper, we conducted a population-based study using data obtained from the National Health Insurance Research Database (NHIRD). Since 1995, Taiwan has started to implement a National Health Insurance (NHI) program that focuses on reducing the financial barrier for all people, in order to obtain sufficient medical services in Taiwan. Until now, the NHI program covers 99.5% of the population [24,25,26]. The NHIRD can be used to record deidentified data from patients or care providers, including medical institutions and physicians. The database we used in this paper contained information on one million beneficiaries randomly sampled from the NHIRD, and the data included ambulatory care expenditures by visit (CD), details of ambulatory care orders (OO), registry for beneficiaries (ID), and medical institution’s basic data files (HOSB), field name and serial file description. Based on the flow chart shown in Figure 2, the case of emergency care with complete medical records could be extracted by lining APPL_TYPE in the file of CD and OO. The total number of cases in the original database was 1,000,000, and 138,713 cases were selected from the ED from 2011 to 2013.

The subjects of this study were emergency medical users in 2012 who were recorded in the same year. Thus, the data excluded patients who visited the emergency department but were determined to remain outside the triage scale and those with an unknown triaged scale. To conduct this study, institutional ethical approval was obtained from the research ethics committee of Taipei City Hospital (TCHIRB-10512105-W). The sample screening process is shown in Figure 2.

## 3. Results

The patient characteristics and health statuses are listed in Table 1. A total of 138,713 patients were included, with 50.39% females and 49.61% males. Age was categorized into three groups: aged 0-18 (8.69%), aged 19–64 (69.65%), and aged ≥ 65 (21.65%). Most patients came from normal households (97.72%), while a small portion of patients belonged to low-income households (2.28%). In total, 108,036 patients (76.44%) resided in municipal cities, followed by 22,177 patients (15.99%) in satellite cities/towns and 10,500 patients (7.57%) in villages. Approximately 14.74% of all patients were hospitalized, and 47.89% received ambulatory care more than 15 times in the previous year. The patient’s CCI scores consisted of scores of 0 (64.15%), scores of 1–2 (26.74%) and scores ≥ 3 (9.11%).

Approximately 52.89% of subjects came from regional hospitals, followed by academic medical centers (28.99%) and district hospitals (18.12%). Most patients were classified as level 3, accounting for 64.28%, followed by level 4 (21.21%) and level 2 (10.45%). Only a small portion of patients were assigned to level 1 (2.43%) and level 5 (1.63%).

Table 2 indicates the results of multivariate logistic regression for LSI, CT, and SC in terms of demographic factors, patient health status and hospital levels. Most patient characteristics were found to be associated with receiving LSI, CT and SC services. Males had higher odds of receiving LSI (AOR = 1.29, *P* < 0.0001), CT (AOR = 1.22, *P* < 0.0001) and SC (AOR = 1.05, *P* = 0.0149) services. This result coincides with the finding of [27].

Patients aged ≥ 65 had the highest odds of receiving LSI (AOR = 8.01, *P* < 0.0001), CT (AOR = 3.23, *P* < 0.0001) and SC (AOR = 1.66, *P* < 0.0001). Patients categorized into low-income households were found to have a significantly higher AOR of 1.29 (*P* = 0.0069) for LSI, but there was no significant association for CT or SC. The AOR of receiving LSI (*P* = 0.0301), CT (*P* = 0.0068) and SC (*P* = 0.0002) for patients living in satellite cities/towns was significantly slightly higher than that for patients living in municipal cities.

Patients without a record of hospitalization in the previous year were significantly less likely to receive LSI (*P* < 0.0001) but more likely to receive CT (*P* = 0.0010) and SC (*P* = 0.0213). The number of instances of ambulatory care in the previous year was not found to have an impact on receiving LSI, CT and SC. The CCI score was found to have a high impact on receiving LSI and CT but no impact on SC. Patients with higher CCI scores (worse health status) were significantly more likely to receive LSI and CT. Hospitals with higher levels were associated with a greater probability for patients to receive LSI (*P* < 0.0001), CT (*P* < 0.0001) and SC (*P* < 0.0001).

Eventually, the triaged level was also found to have significant impacts on receiving LSI, CT and SC. Level 1 (resuscitation) of the triaged level (assigned acuity) had the highest AOR for LSI (AOR = 68.33, *P* < 0.0001), CT (AOR = 2.52, *P* < 0.0001), and SC (AOR = 1.93, *P* < 0.0001), followed by level 2 for LSI (AOR = 7.17, *P* < 0.0001), CT (AOR = 2.23, *P* < 0.0001), and SC (AOR = 1.74, *P* < 0.0001).

## 4. Discussion

This study found that patient characteristics, patient health status, the accredited levels of hospitals and triaged levels may affect the chance of receiving LSI, CT and SC services. Aged and male patients had a greater chance of receiving these three treatments, and patients from low-income households had a greater chance of receiving LSI but not CT or SC. Basically, the choice of treatment determined by physicians followed the same standard of medical care for each patient despite income and demographic factors. Thus, the use of CT and SC was not found to have an association with income. However, the greater chance of being treated by LSI for patients from low-income households implies that their disease is more severe and that it is required to perform LSI. Previous research has found that income is significantly associated with physical and mental health outcomes. For example, Bell [28] found that income played a moderate role in affecting the association between obesity and depression, which results in a high risk for poor health outcomes [29,30]. ED patients from low-income households have worse health status; thus, the performance of immediate LSI is required.

ED patients residing in rural regions (villages or satellite cities/towns) are associated with higher rates of LSI, CI, and SC treatment. Currently, rural regions have aged populations in Taiwan, as younger generations leave villages for municipal cities in order to work. Previous studies have found that the aged tend to have multiple comorbidities and high mortality [31,32,33]; thus, this reasonably explains the higher rate of LSI, CT, and SC treatments for patients residing in rural regions.

The measurement of health status was found to have a significant impact on the treatment of LSI, CT and SC, except for records of ambulatory care. It is intuitively accepted that a worse health status results in a greater need for the emergency treatments of LSI, CT and SC. Higher CCI scores represent a worse status that is associated with worse outcomes due to more comorbidity and increased risks of organ failure. Thus, more aggressive treatment with LSI and CT is recommended.

Patients who received in-patient care in the previous year were found to have a greater chance of receiving LSI but a lower probability for receiving CT and SC. This implies that patients with worse health status are seen as more emergent cases and more often require LSI, including intubation for airway obstruction, needle thoracocentesis, thoracostomy, tourniquets, application of a chest seal, positive-pressure ventilation for ventilatory inadequacy, etc. Since patients had been hospitalized in the previous year, the previous diagnosis had already been recorded. Thus, the use of CT and SC was not required. In this study, the number of visits for ambulatory care was not found to have a significant impact on these treatments. This implies that visits for ambulatory care in the previous year are independent of patient presentation to the ED.

Hospitals with higher accredited levels provided a greater chance for patients to receive emergency treatments of LSI, CT and SC. This suggests that ED patients highly trust the assigned levels of hospitals. District hospitals provided less of a chance for the use of LSI, CT, and SC than regional hospitals and academic medical centers. This may be explained by the limited medical resources in district hospitals, including the availability of CT scanners and quality physicians with sufficient experience in complicated diseases. Academic medical centers in general have higher budgets provided by the government or funds to install expensive equipment such as CT scanners. Table 3 lists the capability of hospitals, including the number of physicians, general ED beds, psychiatric ED beds, and CT scanners according to the accredited level of the hospital. Among these hospitals, academic medical centers had more medical resources than other hospitals. Each academic medical center had 595 physicians, 986 general ED beds, 47 psychiatric ED beds, and 4.23 CT scanners, while each district hospital had 14 physicians, 57 general ED beds, 3.54 psychiatric ED beds, and 0.48 CT scanners. This suggests that hospitals with higher accredited levels equipped with more medical resources and an increased availability of medical resources may motivate the utilization of these resources.

Basically, patients with more severe triaged level have more chance of receiving the treatments of LSI, CT and SC. However, the AOR of Level 4 and 5 for LSI is 0.35 and 0.48, respectively. A patient assigned to lower severity (Level 5) has more chance to receive the treatment of LSI. This may be explained by the fact that patients with thalassemia may receive blood transfusion at emergency departments and the blood transfusion for these patients is probably treated as one of the LSIs in this study. These patients are in general triaged as Level 5 of the triaged level. Thus, the AOR of Level 5 for LSIs indicated in Table 2 may be over-estimated even though the patients with thalassemia account for a small percentage of total samples.

## 5. Limitation

This study was based on the database extracted from National Health Insurance (NHI); thus, the accuracy of the data used may have affected the results. Furthermore, the database was not tailored for this study. The format of the data field was given and thus limits the depth and scope of the study. The data regarding patient wait times for triage and physicians and the total length of stay in the emergency department were not available in the National Health Insurance database. Thus, the analysis of the relationship between patient wait times and the prescribed time for each triaged level was neglected in this study.

## 6. Conclusions

This study found that several patient characteristics, patient health status, the accredited level of medical institutions and the triaged level were associated with a higher likelihood of receiving emergency treatments. The probability of receiving the emergency treatments of LSI, CT and SC was found to be associated with the accredited level of hospitals. Hospitals with higher accredited levels were more likely to provide the emergency treatments of LSI, CT and SC. This study highlights that the inequality of medical resources among medical institutions with different accredited levels may yield a crowding-out effect. Academic medical centers with more capabilities (including physicians and equipment) may attract an increasing number of visiting patients, despite the emergency and severity of diseases.

To avoid the crowding-out effect, administration units should motivate the public to seek general care for nonurgent disease or to access clinics first that conduct a preliminary diagnosis and make a referral to hospitals for in-need patients. These ED treatments (LSI, CT, and SC) may reduce mortality and increase patient satisfaction, but the increase in cost is accompanied by overuse. Some strategies are needed to decrease the overuse of medical resources.

A lack of or delay in the performance of LSI may have a very serious negative effect on mortality. However, clinical judgment regarding the need for immediate LSI in EDs is very difficult to determine, as it is uncertain with regard to the expected outcome [34]. The use of CT may result in some adverse effects and increased costs. The delay in SC may aggravate the overcrowding problem in EDs [35], reduce the quality of care and increase the transfer rates of patients to hospitals with higher accredited levels [36,37]. Thus, a study on the timing for the choice of emergency treatments and the driving force behind the association between emergency treatments and affecting factors may be a focus of future studies.

## Figures and Tables

**Figure 1 ijerph-17-02914-f001:**
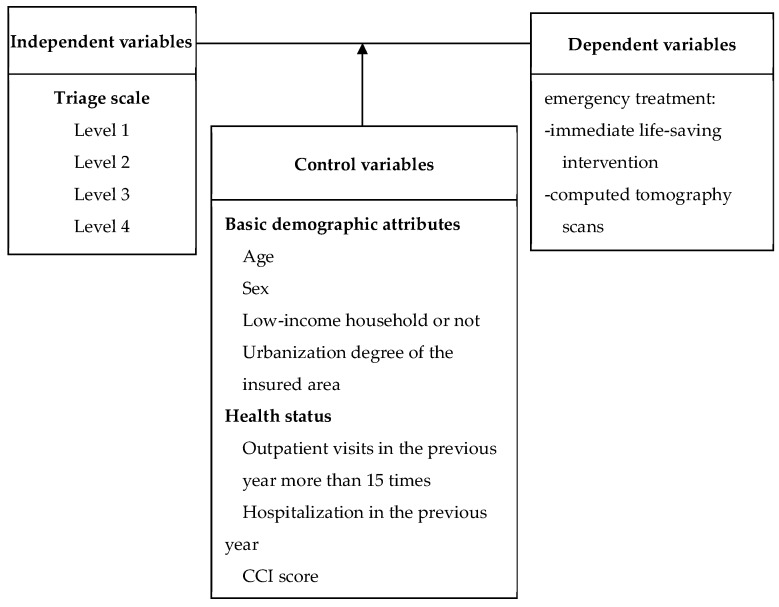
Research structure.

**Figure 2 ijerph-17-02914-f002:**
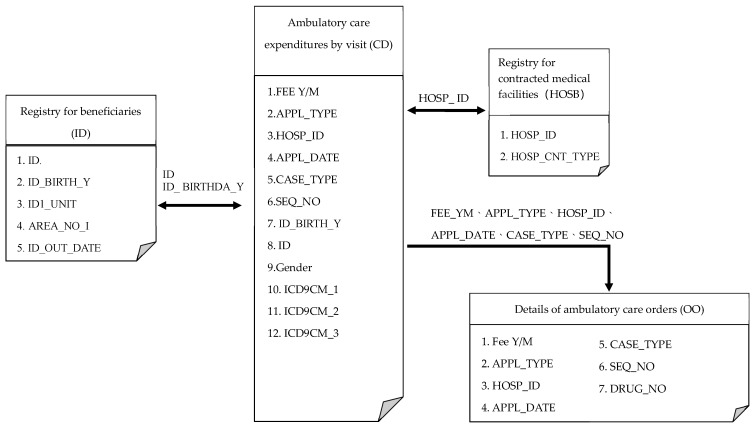
Archives’ name and serial description chart.

**Table 1 ijerph-17-02914-t001:** Descriptive statistics.

Variables	Number of Patients (*n* = 138,713)	%	*p* Value
Basic characteristics			
Patient sex			0.0036
Female	69,898	50.39%	
Male	68,818	49.61%	
Categories by age			<0.0001
0–18	12,060	8.69%	
19–64	96,620	69.65%	
≥ 65 and older	30,033	21.65%	
Low-income households			<0.0001
Yes	3163	2.28%	
No	135,550	97.72%	
Urbanization degree			<0.0001
Municipal city	106,036	76.44%	
Satellite city/town	22,177	15.99%	
Village	10,500	7.57%	
Health status in the previous year			
Hospitalization in the previous year			<0.0001
Yes	20,450	14.74%	
No	118,263	85.26%	
Number of instances of ambulatory care less than 15 times in the previous year			<0.0001
Yes	72,279	52.11%	
No	66,434	47.89%	
CCI (Charlson Comorbidity Index) score			<0.0001
0	88,981	64.15%	
1–2	37,091	26.74%	
≥3	12,641	9.11%	
Level of emergency treatment hospitals			<0.0001
Medical center	40,215	28.99%	
Regional hospital	73,361	52.89%	
Local hospital	25,137	18.12%	
Emergency treatment conditions			
Triaged scale			<0.0001
Level 1	3374	2.43%	
Level 2	14,490	10.45%	
Level 3	89,967	64.28%	
Level 4	29,417	21.21%	
Level 5	2265	1.63%	
Immediate life-saving intervention			<0.0001
Yes	6977	5.03%	
No	131,736	94.97%	
Computed tomography			<0.0001
Yes	12,304	8.87%	
No	126,409	91.13%	
Specialist consultation			<0.0001
Yes	11,353	8.18%	
No	127,360	91.82%	
Hospitalization after emergency treatment			<0.0001
Yes	19,122	13.79%	
No	119,591	86.21%	
In-hospital mortality			<0.0001
Yes	998	0.72%	
No	137,715	99.28%	

**Table 2 ijerph-17-02914-t002:** Factors affecting emergency medical treatment (*n* = 138,713).

Variable Name	Immediate Life-saving Intervention		Computed Tomography		Specialist Consultation	
	AOR (95%CI)	*p* Value	AOR (95%CI)	*p* Value	AOR (95%CI)	*p* Value
Patient- & disease-related factors						
Patient sex						
Female	1		1		1	
Male	1.29 [1.22,1.37]	<0.0001 ***	1.22 [1.18,1.27]	<0.0001 ***	1.05 [1.01, 1.09]	0.0149 *
Patient age						
0–18	1		1		1	
19–64	3.85 [3.06,4.83]	<0.0001 ***	1.81 [ 1.65, 2.00]	<0.0001 ***	1.45 [ 1.33,1.59]	<0.0001 ***
≥65	8.01 [6.35,10.12]	<0.0001 ***	3.23 [ 2.91, 3.58]	<0.0001 ***	1.66 [ 1.51,1.83]	<0.0001 ***
Low-income households						
No	1.29 [1.07,1.55]	0.0069 **	1.06 [ 0.93,1.21]	0.4128	0.91 [ 0.79,1.05]	0.2048
Yes	1		1		1	
Urbanization degree in the insured area						
Urban	1		1		1	
Satellite towns	1.09 [1.01,1.18]	0.0301 *	1.08 [ 1.02,1.14]	0.0068 **	1.12 [ 1.05,1.18]	0.0002 ***
Villages	1.08 [0.98,1.20]	0.1376	1.28 [ 1.19,1.37]	<0.0001 ***	1.22 [ 1.13,1.32]	<0.0001 ***
Medical utilization & health status						
Hospitalized in the previous year						
Yes	1		1		1	
No	0.69 [0.64,0.73]	<.0001 ***	1.10 [ 1.04,1.16]	0.0010 **	0.94 [ 0.88,0.99]	0.0213 *
Less than 15 outpatient visits						
Yes	1		1		1	
No	0.95 [0.88,1.02]	0.1692	1.02 [ 0.97,1.06]	0.5298	0.96 [ 0.91,1.00]	0.0586
CCI score						
0	1		1		1	
1–2	1.43 [1.33,1.54]	<0.0001 ***	1.13 [1.08,1.19]	<0.0001 ***	0.98 [0.93,1.03]	0.3461
≥3	1.77 [1.61,1.95]	<0.0001 ***	1.07 [0.99,1.14]	0.0821	1.02 [0.95,1.10]	0.5559
Health care system-related factors						
Level of emergency hospital						
Medical center	1		1		1	
Regional hospital	0.62 [0.59,0.66]	<0.0001 ***	0.69 [0.66,0.72]	<0.0001 ***	0.40 [0.39,0.42]	<0.0001 ***
Local hospital	0.56 [0.51,0.62]	<0.0001 ***	0.45 [0.42,0.48]	<0.0001 ***	0.12 [0.11,0.13]	<0.0001 ***
Emergency triage scale						
Level 1	68.33 [64.48,74.72]	<0.0001 ***	2.52 [2.31,2.74]	<0.0001 ***	1.93 [1.75,2.12]	<0.0001 ***
Level 2	7.17 [6.72,7.64]	<0.0001 ***	2.23 [2.13,2.34]	<0.0001 ***	1.74 [1.65,1.84]	<0.0001 ***
Level 3	1		1		1	
Level 4	0.35 [0.31,0.41]	<0.0001 ***	0.32 [0.29,0.34]	<0.0001 ***	0.60 [0.57,0.64]	<0.0001 ***
Level 5	0.48 [0.31,0.75]	0.0013 **	0.24 [0.18,0.33]	<0.0001 ***	0.47 [0.37,0.59]	<0.0001 ***

Note: AOR (adjusted odds ratio); CI (confidence interval).

**Table 3 ijerph-17-02914-t003:** The capability of hospitals according to accredited levels by the end of 2017.

Accredited level	Hospitals	Physicians	General ED beds	Psychiatric ED beds	CT
Academic medical centers	22	13,094	21,693	1035	93
Quasi-medical centers	2	839	1299	170	6
Regional hospitals	75	9963	31,440	2144	147
District hospitals	307	4400	17,440	1088	146
Others	77	987	1319	2962	7
Total	483	29,283	73,191	7399	399

Source: MOHW (2018), the status of medical institutions and the quantity of medical services, Ministry of Health and Welfare in 2017.

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
