# Peer review of "Factors Affecting Treatment with Life-Saving Interventions, Computed Tomography Scans and Specialist Consultations"

_ijerph, 2020, doi:10.3390/ijerph17082914_

Round 1
Reviewer 1 Report
I am concerned that the precision of the data used does not reliably reproduce the precision of the data as indicated in limitations, as the results could be affected.
It is not indicated that this study or its study has passed an ethics committee to obtain the data, is this possible ????
Reviewer 2 Report
- There are no references to the results of triage by nurses in the introduction. This can be improved.
- The method does not justify intervening between the ages of 18 and 65. There may be differences in the age range of 45-65 years. In Europe, there is evidence of different use of the health system in this age range.
-
The result does not specify which professional does the triage. There is evidence of the benefits of triage performed by graduate nurses, for example.
On the other hand, the methodology described in Table 2 differs from the results presented. In the method section, 5 types of triage are distinguished. However, in Table 2 triage is grouped into 3 blocks. This is not consistent. The authors should clarify this discrepancy. In the case of grouping the triages, justification should be given.
Reviewer 3 Report
- Line 91: “low-income households by local municipalities if their average monthly income in the household falls below the lowest living index that is based on the living standard and defined by the government.” Please give the exact amount (USD) of low-income households.
- Line 97: Why the definition of worse health status is with “received ambulatory cares more than 15 times in the previous year”
- Please make sure the major six merging factors between DD-DO and CD-OO in Figure 2.
- How many cases in the original database? How many cases were selected into ED in flow chart and Figure 2?
- Please add some statistic methods within “2. Research Methods”.
- As author mentioned in Line 81: “And thus, the emergency treatment serves as a dependent variable while triaged levels are employed as independent variables.” It’s seems “Table 1” should adopt emergency treatments (LSI, CT, and SC) as the title and test for statistic significant.
- Line 133~154: It’s seems something wrong with “Table 2. Over- and under-triage rates triaged by the TTAS”. Please revise Table 2 with the OR and AOR between emergency treatments (LSI, CT, and SC). Furthermore, what are the AOR factors?
- Please add emergency treatments (LSI, CT, and SC) between different accredited levels in Table 3.
